# The Association of Physical Activity and Sedentary Behavior with Maternal and Cord Blood Anti-Oxidative Capacity and HDL Functionality: Findings of DALI Study

**DOI:** 10.3390/antiox12040827

**Published:** 2023-03-28

**Authors:** Saghi Zafaranieh, Julia T. Stadler, Anja Pammer, Gunther Marsche, Mireille N. M. van Poppel, Gernot Desoye, DALI Core Investigator Group

**Affiliations:** 1Institute of Human Movement Science, Sport and Health, University of Graz, 8010 Graz, Austria; 2Division of Pharmacology, Otto Loewi Research Center for Vascular Biology, Immunology and Inflammation, Medical University of Graz, 8010 Graz, Austria; 3Department of Obstetrics and Gynecology, Medical University of Graz, 8036 Graz, Austria

**Keywords:** obesity, pregnancy, antioxidative capacity, physical activity, sedentary behavior, cholesterol efflux

## Abstract

Obesity is one of the most common health issues in pregnancy with short and long-term consequences for both mother and her offspring. Promoting moderate to vigorous physical activity (MVPA) and decreasing sedentary time (ST) could have a positive impact on weight and obesity management, and therefore adiposity-induced oxidative stress, inflammation, and atherogenesis. However, the effects of MVPA and ST on anti-oxidative and anti-atherogenic markers in pregnancy have not been studied to date. This study aimed to assess the association of longitudinally and objectively measured MVPA and ST in 122 overweight/obese women (BMI ≥ 29 kg/m^2^) with maternal and cord blood markers of oxidative stress measured by advanced oxidation protein products (AOPP), anti-oxidative capacity, as well as high-density lipoproteins (HDL) related paraoxonase-1 (PON-1) activity and cholesterol efflux. Linear regression models showed no associations of MVPA and ST with outcomes in maternal blood. In contrast, MVPA at <20 weeks and 24–28 weeks of gestation were positively associated with anti-oxidative capacity, as well as PON-1 activity of HDL in cord blood. MVPA at 35–37 weeks correlated with higher AOPP, as well as higher anti-oxidative capacity. ST <20 weeks was also positively associated with inhibition of oxidation in cord blood. We speculate that increasing MVPA of overweight/obese women during pregnancy attenuates the oxidative stress state in the new-born.

## 1. Introduction

According to World Health Organization (WHO), the prevalence of obesity is rising worldwide and has reached pandemic levels, also among pregnant women [1]. Obesity is a major risk factor for many maternal complications, including preeclampsia, hypertension, and gestational diabetes. Children born to women with high body mass index (BMI) are also prone to cardiovascular and metabolic complications later in life [2].

One of the reasons for the negative influence of obesity on maternal and offspring outcomes, might be in the fact that obesity promotes the generation of free radicals, causing oxidative and nitrative stress [3]. Reactive oxygen species (ROS) are hyper-reactive molecules. At physiological levels, they are involved in cellular signaling pathways and are important for proper cellular function. However, ROS excess may result in lipid peroxidation, protein modifications, and DNA oxidation, with the potential to induce cellular damage and impact tissue function [4]. Therefore, the anti-oxidative capacity of tissues and body fluids in scavenging excess amounts of free radicals is an important parameter in cell signaling and survival.

Moreover, among many consequences of obesity-induced dyslipidemia, altered high-density lipoprotein (HDL) levels and subclass distribution, composition, and function have attracted research interest in recent years. HDL particles have advantageous physiologically relevant characteristics. For example, cholesterol efflux is an anti-atherogenic functionality of HDL to promote reverse cholesterol transport from peripheral cells to the liver for excretion [5]. Additionally, HDL-associated enzymes, such as paraoxonase-1 (PON-1) share anti-oxidative and anti-inflammatory properties by protecting low-density lipoproteins (LDLs) from oxidation [6]. On the other hand, obesity is inversely correlated with HDL cholesterol levels and impairs HDL function [7,8].

Healthy lifestyle behaviors are essential in the prevention of obesity and other chronic conditions, and may also mitigate the negative consequences of obesity on maternal oxidative stress and HDL function during pregnancy. There is growing evidence that physical activity (PA) during pregnancy contributes positively to pregnancy and birth outcomes and reduces the risk of obstetric complications [9]. The American College of Obstetrics and Gynecologists (ACOG) recommends all women without contraindications to participate in at least 150 min of moderate PA per week [10]. Increasing and maintaining physical activity and reducing sedentary behavior could control gestational weight gain, and consequently obesity and dyslipidemia [11,12,13]. PA boosts anti-oxidative capacity and ameliorates immuno-metabolic health outside of pregnancy [14,15]. However, the role of PA in pregnancy in altering/affecting maternal oxidative stress is not clear. Furthermore, how the metabolic environment of the mother is conveyed to the fetus remains unclear. For instance, how maternal hypercholesterolemia and hyperglycemia are associated with changes in serum lipid concentrations, lipoprotein composition, and oxidative stress in the fetus [16,17]. Therefore, there is reason to hypothesize an influence of maternal PA on fetal oxidative stress, but this has not been studied to date.

This study sought to explore the association between objectively measured PA and sedentary time (ST) during pregnancy with maternal and cord blood levels of advanced oxidation protein products (AOPPs), an established marker of oxidative stress, serum anti-oxidative capacity, and essential anti-atherogenic HDL functions, such as the ability to remove cholesterol from macrophages (cholesterol efflux capacity), and the activity of the HDL-associated anti-inflammatory enzyme PON-1.

## 2. Materials and Methods

### 2.1. DALI Study

This is a secondary analysis of the vitamin D and lifestyle intervention for GDM prevention (DALI) randomized controlled trial, which was carried out between 2012 and 2015 in eleven study centers across nine different European nations: Austria, Belgium, Denmark (Odense, Copenhagen), Ireland, Italy (Pisa, Padua), The Netherlands, Poland, Spain, and the United Kingdom. The study received approval from all regional ethics committees and was registered under trial registration number ISRCTN70595832. [18]. The sub-study presented here was approved by the ethics committee of the Medical University of Graz (number 30-486 ex17/18).

Pregnant women with a gestational age of <20 weeks and a singleton pregnancy, aged ≥18 years, and with a pre-pregnancy BMI of ≥29 kg/m^2^ were invited to participate. Women who had gestational diabetes mellitus (GDM) at baseline according to International Association of Diabetes and Pregnancy Study Group (IADPSG) criteria or who had been previously diagnosed with diabetes, chronic medical conditions, or mental disease were excluded. Additional exclusion criteria were: The inability to walk 100 m safely, a need for a complex diet, communication difficulties with the lifestyle coach owing to language barriers. Following the written informed consent, women were randomly divided into four groups that were consulted in Healthy Eating (HE), Physical Activity (PA), Healthy Eating + Physical Activity (HE + PA) and the control group receiving usual care (UC). For this analysis, all randomized participants were combined into one cohort.

### 2.2. Data Collection

Participants’ data and blood samples were collected at four time-points: Baseline (<20 weeks), 24–28 weeks, 35–37 weeks, and after delivery. Maternal data included weight, height, age, ethnicity, parity, pre-pregnancy weight, maternal/paternal smoking, alcohol consumption, and medical history. Venous cord blood was collected immediately after birth. Data on birth outcomes were collected from medical files. Maternal and cord blood were used to determine the anti-oxidative capacity and anti-atherogenic functional capacity of HDL.

### 2.3. Physical Activity and Sedentary Time

As previously described [19], physical activity was measured objectively by accelerometers (ActiGraph GT1M, GT3X+ or Actitrainer; Pensacola, FL, USA) at <20 weeks, 24–28 weeks, and 35–37 weeks. A minimum of three valid full day measurements per time-period, with a daily wear time of more than 480 min, were needed to be included in the analysis.

The average time per day spent sedentary (<100 counts/min), light (100–1951 counts/min), and in moderate-to-vigorous physical activity (MVPA) (>1951 counts/min) was determined [20]. Swimming time was added to MVPA time [21]. Sedentary time was determined as a proportion of total daily accelerometer wear time (%ST) for analyses, since changes in wear time influence the absolute number of minutes of sedentary time considerably. Due to multicollinearity, light physical activity was left out of the model, as together with sedentary time and MVPA it accounts for 100% of the measured daily physical activity.

### 2.4. ApoB-Depletion of Serum

The HDL composition and function were assessed using apoB-depleted serum. To 100 µL serum, 40 µL of polyethylene glycol (Sigma Aldrich, Darmstadt, Germany) (20% in 200 mmol/L glycine buffer) was added, mixed gently, and then incubated for 20 min at room temperature. After a centrifugation step at 10,000 rpm for 30 min at 4 °C, the supernatant was collected. Samples were stored at −70 °C until usage.

### 2.5. Advanced Oxidation Protein Products (AOPP)

AOPPs were assessed with slight modifications of previously described protocols [22]. To 10 μL apoB-depleted serum, 40 μL 0.2 mol/L citrate buffer was added, mixed, and incubated for 2 min on a shaker. Thereafter, absorbance was measured at 340 nm. AOPP was calibrated with chloramine-T (linear within the range of 0 to 100 μmol/L) and were expressed as μmol/L of chloramine-T equivalents.

### 2.6. Anti-Oxidative Capacity of apoB-Depleted Serum

The anti-oxidative activity of apoB-depleted serum was determined using the fluorescent dye dihydrorhodamine as in the previously described method with modifications [23]. Briefly, dihydrorhodamine was suspended in DMSO to a 50 mM stock, which was diluted in HEPES (20 mM HEPES, 150 mM NaCl, pH 7.4) containing 1 mM 2,2′-azobis-2-methyl-propanimidamide-dihydrochloride (Sigma-Aldrich, Darmstadt, Germany) to a 10 μM working reagent. Ten μL of apoB-depleted serum (1:10 diluted) were placed in 384-well plates (Greiner Bio-One, Kremsmünster, Austria) and the volume was adjusted to 100 μL with HEPES buffer containing 10 μM dihydrorhodamine. The increase in fluorescence due to the oxidation of dihydrorhodamine was monitored using an xMark plate reader (Biorad, Vienna, Austria) for 90 min at 538 nm. The rate of oxidation became linear after an initial lag phase of about 20 min. The increase in fluorescence was calculated from the linear range, and then was used as a benchmark. The increase in dihydrorhodamine fluorescence per minute in the absence of apoB-depleted serum or isolated HDL was set to 100%, and individual apoB-depleted serum (three independent experiments measured in duplicates) or isolated HDL (three independent experiments each measured in duplicates) samples were calculated as a percentage of inhibition of dihydrorhodamine oxidation.

### 2.7. Arylesterase Activity of Paraoxonase-1

Ca2+-dependent arylesterase activity of paraoxonase-1 (PON-1) in apoB-depleted serum was assessed using a photometric assay with phenylacetate substrate [24]. Activities were calculated from the slopes of the kinetic chart of four independent experiments, measured in duplicates.

### 2.8. Cholesterol Efflux Capacity

The cholesterol efflux capacity of apoB-depleted serum was assessed, as described elsewhere [25,26]. Briefly, J774.2 cells (Sigma Aldrich, Darmstadt, Germany) were cultured in DMEM medium (Life Technologies, Carlsbad, CA, USA) containing 10% FBS and 1% penicillin/streptomycin. In 48-well plates, a total of 300,000 cells per well were seeded and cultured for 24 h. Subsequently, cells were labeled with 0.5 µCi/mL radiolabeled [3H]-cholesterol (Hartmann Analytic, Braunschweig, Germany) in DMEM media containing 2% FBS, 1% penicillin/streptomycin, and 8(4-chlorophenylthio)-cyclic adenosine monophosphate (0.3 mM) (Sigma-Aldrich, Darmstadt, Germany) overnight. After two rounds of rinsing, cells were equilibrated for 2 h with serum-free DMEM containing 2% bovine serum albumin (Sigma-Aldrich, Darmstadt, Germany). After rinsing, the sample containing 2.8% apoB-depleted serum was added and incubated for 3 h. Cholesterol efflux capacity was expressed as a ratio of radioactivity in the media to the total radioactivity of media and lysed cells.

### 2.9. Statistical Methods

Study participants’ characteristics are presented by mean and standard deviation (SD) or count and proportion. MVPA and %ST levels are presented for each time point and the differences between time points were tested by ANOVA or Wilcoxon test. The differences in anti/pro-oxidative and HDL-related parameters were tested using ANOVA and pairwise comparisons based on estimated marginal means or Friedman and Wilcoxon test.

To analyze the association between measured MVPA and %ST at <20 weeks, 24–28 weeks, and 35–37 weeks of gestation with correspondent measurement of maternal blood markers at those time-points, as well as with cord blood markers, multi-variable linear regression models were used. To be able to estimate associations independently of one another, the MVPA and %ST variables were included in the same model. The models were additionally adjusted for maternal age (continuous), parity (null parity vs. multiparity), and pre-pregnancy BMI (continuous). Adjustments for the mode of delivery (spontaneous vs. elective), Caesarean section (yes vs. no), healthy eating intervention (yes vs. no), GDM (yes vs. no), or study center did not change the associations, and therefore, were left out of the model due to the relatively small sample size.

In regard to sensitivity analysis, smoking participants were excluded to inspect whether this variable influenced the relationship between maternal MVPA and %ST and the laboratory blood outcomes.

Descriptive analysis and the main analysis of the study were conducted with IBM SPSS Statistics (version 27.0). Plots were produced with GraphPad Prism software (version 9.2.0).

## 3. Results

### 3.1. Study Participants’ Characteristics

Information on maternal blood parameters was available for 182 DALI participants, and 122 women had accelerometer measurement from at least one time-point and were included in the analysis. Characteristics of the participants are shown in Table 1. Most women were highly educated, non-smokers, and Caucasian, living with their husband or partner. Women had an average age of 32.8 ± 5.2 years and a pre-pregnancy BMI of 33.7 ± 4.1 kg/m^2^. Comparison of included and excluded women is given in Appendix A. Included women were somewhat older, higher educated, less frequently developed GDM, and birthweight of their offspring was higher, compared to excluded women.

### 3.2. Physical Activity and Sedentary Time during the Course of Pregnancy

Average MVPA and %ST at three different stages of pregnancy are summarized in Table 2. In this sample of pregnant women, MVPA significantly decreased from <20 weeks to 35–37 weeks (*p* = 0.004), whereas %ST increased over time, but not significantly.

### 3.3. Markers of Oxidative Stress and HDL Functional Parameters in Maternal and Cord Blood

Advanced oxidation protein products (AOPPs) decreased from <20 weeks to 24–28 weeks (median (IQR) of 21.8 (14.6) vs. 20.9 (10.4)) and did not change toward late pregnancy (21.3 (14.9) (Figure 1A). At all three time points, AOPPs in maternal blood were lower than in cord blood (median 38.9.1 (IQR 32.4); all *p* < 0.0001).

Changes in maternal serum anti-oxidative capacity, PON-1-related AE activity, and cholesterol efflux capacity in pregnancy were similar as previously reported in a sample of 192 women in the DALI study [27]. The serum anti-oxidative capacity (% inhibition of oxidation) increased significantly from early to late pregnancy in maternal serum, from 56.6% ± 5.8 at <20 weeks, 57.1% ± 5.8 at 24–28 weeks, to 58.7% ± 5.6 at 35–37 weeks (Figure 1B). In cord blood, anti-oxidative capacity was 70.6% ± 7.3 unit, and this was greater than in maternal blood at all three time points of gestation (all *p* < 0.0001).

PON-1-related AE activity did not change in maternal blood between different time points (Figure 1C) and was significantly lower in cord blood (38.1 (44.7) mM/min/mL) compared to all time points (<20 weeks: 185.1 (71.7), 24–28 weeks: 174.5 (66.4) and 35–37 weeks: 179 (55.8)) in maternal blood (all *p* < 0.0001).

Maternal average cholesterol efflux capacity was significantly higher at late gestation (14.5% ± 2.4) compared to <20 weeks (13.5% ± 2.3; *p* < 0.001) and 24–28 weeks (13.9% ± 2.2; *p* = 0.01) (Figure 1D). Cord blood cholesterol efflux capacity (7.4% (2.7)), however, was lower compared to maternal blood (13.2% (3.4), 13.7% (3.1), 14.2% (2.7), respectively; *p* < 0.0001). However, after normalization to HDL-C levels, the cholesterol efflux capacity of individual HDL particles in cord blood (median (IQR): 13.8 (5.5)) was significantly higher than in maternal blood (<20 weeks: 8.9 (2.7), 24–28 weeks: 9.1 (2.8) and 35–37 weeks: 9.5 (3); all *p* < 0.001).

### 3.4. Maternal MVPA or ST and Pro/Anti-Oxidative Stress and HDL-Related Markers in Maternal Blood

We used a linear regression model adjusted for maternal age, pre-pregnancy BMI, and parity to investigate the relationship between MVPA and %ST at <20 weeks, 24–28 weeks, and 35–37 weeks of gestation with correspondent measurements of maternal blood markers. No associations were found between maternal levels of AOPP, anti-oxidative capacity measured as % inhibition of oxidation, AE activity of PON-1, or cholesterol efflux capacity, with measured MVPA and %ST of the women (Figure 2). Additional adjustment for HE intervention, mode of delivery, Caesarean section, or GDM did not change the results.

### 3.5. Association of MVPA and ST with Oxidative Stress Markers and HDL-Related Functionalities in Cord Blood

Accelerometer data and cord blood samples were available from 61 pregnancies. The association of MVPA and %ST at three time points during pregnancy with anti and pro-oxidative, as well as HDL-related anti-atherogenic markers in cord blood are presented in Appendix A.

Higher MVPA levels at all three time points during pregnancy were associated with higher anti-oxidative capacity in cord blood (Figure 3). AOPP levels were positively associated with MVPA only in late pregnancy. Moreover, PON-1 activity in cord blood increased with higher MVPA at <20 weeks and 24–28 weeks of gestation. MVPA was not associated with HDL cholesterol efflux capacity.

Higher levels of %ST at <20 weeks of gestation were positively associated with serum anti-oxidative capacity in cord blood, whereas AOPPs, AE activity of PON-1, and cholesterol efflux capacity were not associated with %ST (Figure 3).

Subsequently, the interactions of each association with neonatal sex were investigated and no interactions were found. Additional adjustment for HE intervention, mode of delivery, Caesarean section, or GDM did not change the associations.

### 3.6. Sensitivity Analyses

In regard to sensitivity analyses, smoking participants (n = 12) were excluded to test for possible influences of smoking on the relationship between maternal MVPA and ST and blood markers. Estimates and statistical significance of the sensitivity analyses in non-smoking women were similar to the main analyses, except for the association of MVPA and %ST with cord blood AE activity. In non-smoking participants, the estimate was smaller and not significant anymore, probably due to the smaller sample size (Appendix A).

## 4. Discussion

The main findings of the present study are: (1) Maternal MVPA or ST did not influence maternal serum anti-oxidative capacity and HDL-related anti-atherogenic functionalities in obese pregnant women; (2) in women, who had higher levels of MVPA in pregnancy, higher serum anti-oxidative capacity, AE activity of PON-1, and higher levels of AOPP were measured in cord blood; (3) no association of MVPA or ST was found with HDL cholesterol efflux capacity in cord blood.

Serum anti-oxidative capacity in cord blood is decreased in women with obesity compared to non-obese women [17]. Physical activity is important to mitigate this negative effect of obesity. We recently reported that increasing MVPA and reducing ST attenuate the oxidative stress state in the placentas of obese pregnant women [19]. In the present study, we observed that MVPA is associated with higher antioxidant capacity in cord blood. At the same time, MVPA is associated with an increase in the level of AOPPs, a reliable marker for oxidant-mediated protein damage and oxidative stress [28,29]. Collectively, our results suggest that the reduction in free radicals by the increase in anti-oxidative capacity may be a compensatory mechanism in response to the increased oxidative stress induced by MVPA. This compensatory mechanism has also been reported outside of pregnancy. Physical activity induces an acute increase in oxidative stress and an increase in LDL oxidation. However, in the long term, repeated PA results in upregulation of antioxidative and anti-inflammatory processes [30,31].

Therefore, the absence of association between MVPA with oxidative stress or HDL function in maternal serum is surprising. An explanation could be that in pregnant women with obesity, their pre-pregnancy BMI and hyperglycemia override a possible positive compensatory influence of PA on oxidative stress and inflammation [32]. Obesity could induce systemic inflammation, as well as the release of lipid peroxides and ROS into the maternal circulation [33]. Although the obese women included in the study on average met the PA recommendation for pregnancy, their PA may not have been of sufficient intensity or duration to induce changes in maternal blood. Another explanation could be that pregnancy itself prevented an association of MVPA and ST with oxidative stress in maternal blood. Inflammatory and oxidative stress markers are closely interrelated, and pregnancy itself is an inflammatory state, which recuperates after delivery [34,35].

Fetal and maternal factors both have an impact on the levels and composition of lipids and lipoproteins in cord blood. Owing to its associated proteins and enzymes, HDL has unique anti-oxidative activity. HDL-associated PON-1 activity contributes to its anti-oxidative and anti-atherogenic activity by inhibiting the formation of oxidized LDL and disabling LDL-derived oxidized phospholipids after formation [6]. HDL’s role in reverse cholesterol transport has also been established in maternal and fetal circulation and is inversely associated with cardiovascular events independent from HDL-C level [36,37].

The contrast between cord blood and maternal blood is striking. The association of MVPA with antioxidative capacity and PON-1 activity in cord blood was not paralleled by similar results in maternal blood. Fetal and neonatal HDL differ from maternal HDL with respect to its proteome, size, and function [38]. Consistent with other studies, HDL-associated PON-1 activity was lower in fetal than in maternal circulation [34]. Of note, here PON-1 activity was higher in cord blood of women who had higher MVPA. PON-1 activity is modulated by genetic factors, polymorphisms, as well as environmental factors, such as smoking, diet, physical activity, age, and disease conditions. Factors that reduce oxidative stress may enhance PON-1 activity [39]. It has been demonstrated that PA boosts PON-1 activity [40]. This differs from our results, which are in line with some, but not all, findings in obese people outside of pregnancy [41,42]. As independent PON-1 predictors, TBARS, leptin, and adiponectin levels demonstrate the significance of obesity in the control of PON-1 [43]. Therefore, a more extensive lifestyle intervention may be required to have notable effects on its activity in obese pregnant women [44].

The capacity of HDL to promote cholesterol efflux can be affected by oxidation, lipolysis, and proteolysis, and is enhanced as a compensatory mechanism in response to diabetes-induced oxidative stress and insulin in the fetal circulation [45]. However, in our study, similar to maternal blood, no associations were seen between maternal PA or ST with cholesterol efflux in cord blood.

### Strengths and Weaknesses

To the best of our knowledge, this is the first study on the relationship between physical activity and sedentary behavior in pregnancy and serum antioxidative capacity and HDL-related anti-atherogenic properties of maternal and cord blood. One major strength of the study is the objective measurement of PA and ST. Self-administered questionnaires, which are frequently used to semi-quantify PA and ST in pregnancy, are prone to bias [46,47]. The study’s prospective and longitudinal design, which included measurements of maternal blood parameters, PA and ST levels at various time points in pregnancy, allowed for testing timing effects of lifestyle-related factors on the outcomes. We included only a subgroup of the total DALI participants in this study, with some small differences between included and excluded women. Nevertheless, the pan-EU nature of the DALI study increases the representativeness of the results for obese Caucasian pregnant women in Europe. Limitations of our study are the small sample size and lack of inclusion of lean pregnant women. Therefore, larger studies including lean women are needed to confirm our results.

## 5. Conclusions

Moderate to vigorous physical activity of overweight/obese pregnant women is positively associated with cord blood anti-oxidative capacity, as well as HDL functionality. This may constitute one mechanism to protect the fetus from oxidative stress associated with maternal adiposity.

## Figures and Tables

**Figure 1 antioxidants-12-00827-f001:**
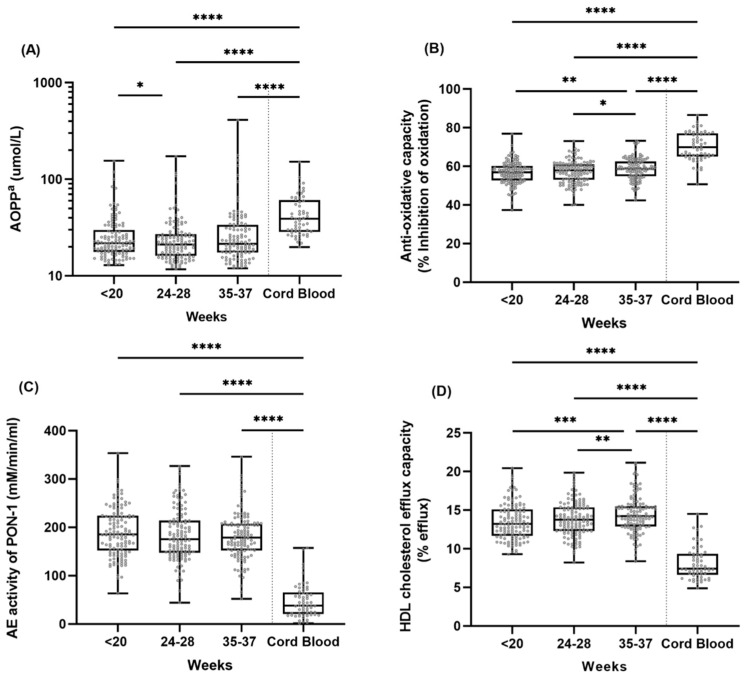
Advanced oxidation protein products (AOPPs) (**A**), % inhibition of oxidation (**B**), AE activity (**C**), and cholesterol efflux (**D**), at three time points during pregnancy in maternal blood, and at term in cord blood. ^a^ Axis values logarithmically scaled. * *p* < 0.05, ** *p* < 0.01, *** *p* < 0.001, **** *p* < 0.0001.

**Figure 2 antioxidants-12-00827-f002:**
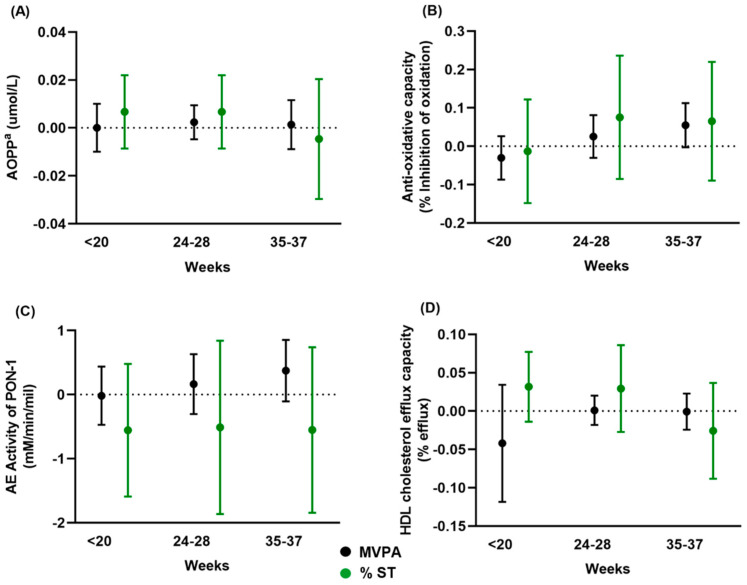
Regression coefficient estimates with 95% confidence intervals of the association between moderate-to-vigorous physical activity (MVPA= black) and % sedentary time (%ST = green) and maternal serum levels of (**A**) advanced oxidation protein products (AOPP), (**B**) serum anti-oxidative capacity (% inhibition of oxidation), (**C**) AE activity of PON-1, and (**D**) HDL cholesterol efflux capacity. ^a^ Log transformed values were used in the regression analyses.

**Figure 3 antioxidants-12-00827-f003:**
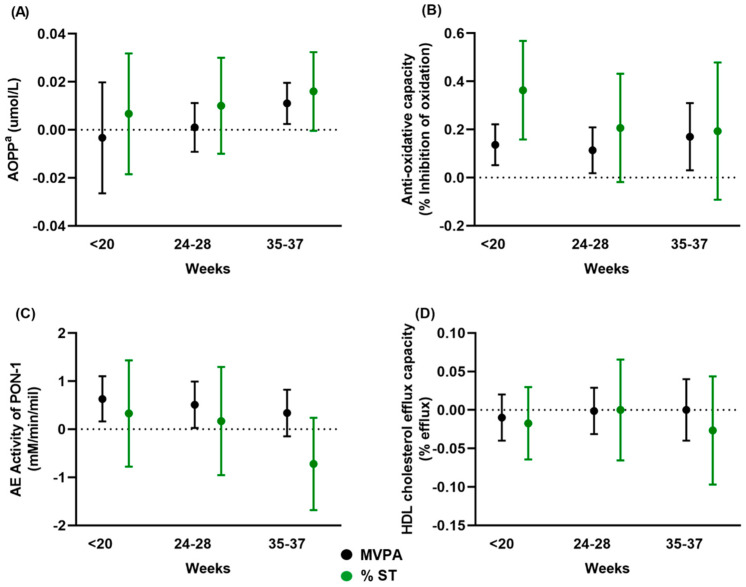
Regression coefficient estimates with 95% confidence intervals of the association between moderate-to-vigorous physical activity (MVPA= black) and % sedentary time (%ST = green) and cord blood levels of advanced oxidation protein products (AOPP) (**A**), anti-oxidative capacity (% inhibition of oxidation) (**B**), AE activity of PON-1 (**C**), and HDL cholesterol efflux capacity (**D**). ^a^ Log transformed values were used in the regression analyses.

**Table 1 antioxidants-12-00827-t001:** Characteristics of the study sample.

Maternal Characteristics	Total N = 122
Age, years, mean ± SD	32.8 ± 5.2
Pre-pregnancy BMI, kg/m^2^, mean ± SD	33.7 ± 4.1
Gestational weight gain, kg, mean ± SD	8.2 ± 4.9
Primiparous, n (%)	55 (45%)
Married/living with a partner, n (%)	114 (93%)
High education, n (%)	80 (66%)
European descent, n (%)	108 (89%)
Smoking, n (%)	12 (10%)
GDM, n (%)	27 (22%)
Caesarean Section, n (%)	33 (27%)
Neonatal characteristics	N = 122
Gestational age at birth, week, mean ± SD	39.8 ± 1.3
Birth weight, g, mean ± SD	3592 ± 480
Female sex, n (%)	57 (47%)

**Table 2 antioxidants-12-00827-t002:** Moderate-to-vigorous physical activity (MVPA) and sedentary time at three time points in pregnancy.

	<20 Weeks N = 86	24–28 Weeks N = 75	35–37 Weeks N = 72
MVPA, min/day, median (IQR)	31.8 (27.1)	29.7 (30.2)	22.0 (27.9) *
Sedentary time, % of wear time, mean ± SD	69.0 ± 10.4	69.8 ± 9.6	71.9 ± 9.5

* *p* = 0.004. MVPA = moderate-to-vigorous physical activity, IQR = interquartile range, and SD = standard deviation.

## Data Availability

The raw data supporting the conclusions of this manuscript will be made available by the authors, without undue reservation, on request to the corresponding author.

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
