# Peer review of "The Association of Physical Activity and Sedentary Behavior with Maternal and Cord Blood Anti-Oxidative Capacity and HDL Functionality: Findings of DALI Study"

_antioxidants, 2023, doi:10.3390/antiox12040827_

Round 1

Reviewer 1 Report

The submitted study investigated the association of physical activity and sedentary behavior with maternal and cord blood antioxidative capacity and HDL functionality.  The study enrolled 122 overweight/obese pregnant women (BMI>29 kg/m2), and assessed longitudinally moderate to vigorous physical activity and sedentary time of the recruited women together with maternal and cord blood markers of oxidative stress.  The results reported in the manuscript indicate that no associations of MVPA and ST with outcomes in maternal blood. A couple of possible explanations were put forward by the authors to justify with rather surprising result. In contrast, MVPA and ST were positively associated with antioxidative capacity and paraoxonase-1 activity of HDL in cord blood. MVPA at 35-37 weeks showed the higher correlation with advanced oxidation protein products and anti-oxidative capacity.  The study was carried out across 9 different European countries and included 11 different centers. 

The article is well written and can be followed easily.

General Comments

 The study does not report whether some European countries were more represented than others in the pool of 122 women recruited for the study, and in the pool of 463 women who were excluded from the study based on preexisting  adverse conditions. The reason for this comment is that different diets, genetic conditions, and gestational cares across the different countries included in the study could have generated small but important differences in the outcome of the study. In the absence of this set of information, it would have been helpful if the authors had provided some of the guidelines established for the study to render the differential recruitment more homogeneous and consequently export the obtained results to much larger populations. 

Results interpretation: 

The results reported in the study are certainly interesting. However, it would have been far more interesting, and appropriate, to compare the data based on: 1) the presence or absence of GDM, which occurred to some level of severity (undefined here) in 22% of the recruited women (Table 1). GDM does result in an increase in systemic inflammation, and this condition may be responsible for some of the large confidence intervals reported in figures 2 and 3, for both MVPA and ST, and 2) natural delivery or delivery by C-section, which occurred in 27% of the women recruited here (also Table 1) 

Reviewer 2 Report

The authors describe that this study investigated the effects of MVPA and ST on antioxidant and anti-atherosclerotic markers, as a follow-up to the authors' recent report that increased moderate to vigorous physical activity (MVPA) and reduced sedentary time (ST) attenuates the oxidative stress state in the placenta of obese pregnant women.

The authors show MVPA is related to the high antioxidant capacity of cord blood in obese pregnant women, thus, suggest that increasing MVPA during pregnancy might attenuate the oxidative stress state in the newborn. They are reporting: (1) Maternal MVPA or ST did not influence serum anti-oxidative capacity and HDL-related anti-atherogenic functionalities of maternal blood; (2) in women, who had higher levels of MVPA in pregnancy, higher serum anti-oxidative capacity, AE activity of PON-1 and higher levels of AOPP were measured in cord blood; (3) No association of MVPA or ST was found with HDL cholesterol efflux capacity in cord blood. Finally, the conclusion is that moderate to vigorous physical activity is positively associated with cord blood anti-oxidative capacity as well as HDL functionality.

The investigation throughout the MS is an analysis of the effect of MVPA and ST during gestation in obese patients, but there is no comparison to non-obese patients for the control. This weakness is more pronounced on line 313 while explaining the results [Maternal MVPA or ST did not affect serum antioxidant capacity and HDL-related anti-atherosclerotic function of maternal blood]. Although the [Strength and weaknesses] section described these weak points, the authors' arguments might be not fully supported.

Reviewer 3 Report

The authors identified and quantified LMs profiles in the serum of patients with CRC by LC-MS/MS. Characterizing patients by age and BMI ,each of these characteristics may affect fatty acid composition it would be appropriate to stratify the results according to these characteristics.

     1) Also specify in materials and methods the number of study participants.

     2) In the table 2 the number of subjects analyzed is describe, specify if all the markers were analyzed on the same number.

Author Response

The reviewer’s comments are somewhat confusing, but we checked whether the number of participants were indeed missing. The number of included participants is mentioned in results (page 5, line 201-202, and page 7, line 264) and in both tables. Therefore, we did not make any changes in the manuscript.

Round 2

Reviewer 2 Report

Thank you for the author's responses. The authors have revised the issues raised by the reviewer.